# Detection of Phytoplankton Temporal Anomalies Based on Satellite Inherent Optical Properties: A Tool for Monitoring Phytoplankton Blooms

**DOI:** 10.3390/s19153339

**Published:** 2019-07-30

**Authors:** Jesús Antonio Aguilar-Maldonado, Eduardo Santamaría-del-Ángel, Adriana Gonzalez-Silvera, María Teresa Sebastiá-Frasquet

**Affiliations:** 1Facultad de Ciencias Marinas, Universidad Autónoma de Baja California, Ensenada 22860, Mexico; 2Alumni PhD Postgraduate Program in Coastal Oceanography FCM-UABC, Ensenada 22860, Mexico; 3Institut d’Investigació per a la Gestió Integrada de Zones Costaneres, Universitat Politècnica de València, 46730 Grau de Gandia, Spain

**Keywords:** remote sensing, absorption coefficients, phytoplankton bloom, MODIS-Aqua, Pacific Ocean, baseline

## Abstract

The baseline of a specific variable defines the average behavior of that variable and it must be built from long data series that represent its spatial and temporal variability. In coastal and marine waters, phytoplankton can produce blooms characterized by a wide range of total cells number or chlorophyll a concentration. Classifying a phytoplankton abundance increase as a bloom depends on the species, the study area and the season. The objective of this study was to define the baseline of satellite absorption coefficients in Todos Santos Bay (Baja California, Mexico) to determine the presence of phytoplankton blooms based on the satellite inherent optical properties index (satellite IOP index). Two field points were selected according to historical bloom reports. To build the baseline, the data of phytoplankton absorption coefficients ($a_{phy{,}_{GIOP}}$) and detritus plus colored dissolved organic matter (CDOM) ($a_{dCDOM{,}_{GIOP}}$) from the generalized inherent optical property (GIOP) satellite model of the NASA moderate resolution imaging spectroradiometer (MODIS-Aqua) sensor was studied for the period 2003 to 2016. Field data taken during a phytoplankton bloom event on June 2017 was used to validate the use of satellite products. The association between field and satellite data had a significant positive correlation. The satellite baseline detected a trend change from high values to low values of the satellite IOP index since 2010. Improved wastewater treatment to waters discharged into the Bay, and increased aquaculture of filter-feeding mollusks could have been the cause. The methodology proposed in this study can be a supplementary tool for permanent in situ monitoring programs. This methodology offers several advantages: A complete spatial coverage of the specific coastal area under study, appropriate temporal resolution and a tool for building an objective baseline to detect deviation from average conditions during phytoplankton bloom events.

## 1. Introduction

Blooms are proliferation events of phytoplankton species, such as dinoflagellates, diatoms and cyanobacteria in aquatic ecosystems [1]. They can last less than 24 h (fast blooms), can last for several days [2] or last for weeks [3]. To be able to detect a bloom, it is necessary to determine the baseline condition of any phytoplankton property in order to detect deviations from that baseline having into account climatic variability. The baseline can be defined as the average value over which fluctuations can occur within a range that contains most of the observed cases [4,5]. The baseline can be calculated from a compilation of years of historical data that define an average of past performance, or from a rapid measurement of current production before initiating a change [6].

Data can be obtained from in situ monitoring, but, in recent years, remote sensing technologies have been indicated as an adequate tool for providing a synoptic view of extensive ocean or coastal areas, being effective in complementing in situ sampling programs [7,8,9]. Chlorophyll a (Chla) has been widely used as a proxy for phytoplankton biomass in both remote sensing and in situ monitoring programs [8]. However, other phytoplankton properties can be monitored from remote sensing and be applied to the study of blooms. For example, Blondeau-Patissier et al. (2014) [10] defined a phytoplankton bloom as “a biological event composed of micro-algal species that is sustained both over time and space and that results in noticeable changes in satellite radiances at wavelengths used for algal bloom proxies due to an increase in biomass in comparison to surrounding algal bloom-free waters”. The inherent optical properties (IOPs) are considered as one of the most robust phytoplankton properties that can be monitored from remote sensing data [10,11,12,13]. The study of IOPs is especially relevant in optically complex waters, such as coastal waters, where optically active constituents (colored dissolved organic matter (CDOM) and detritus) are as important as Chla and show their own patterns during algal bloom conditions [10,14]. Recent findings propose the use of IOPs for the detection of algal blooms [15,16,17], instead of only Chla. The use of IOPs from remote sensing technology may allow long-time monitoring at moderate- or high-spatial resolution and high-temporal resolution [18,19,20], which is essential to build the baseline.

The IOPs are represented by the light absorption coefficients by phytoplankton ($a_{phy}\left(\lambda \right)$), detritus ($a_{d}\left(\lambda \right)$), CDOM ($a_{CDOM}\left(\lambda \right)$) and by the particulate backscattering coefficient ($b_{bp}\left(\lambda \right)$) [21]. A variety of semi-analytical approaches have been proposed to determine IOPs from the remotely measured spectral reflectance, especially for optically complex waters [22,23]. Among them, the algorithm generalized inherent optical property (GIOP) is part of the standard NASA OBPG (Ocean Biology Processing Group)products, with the consideration that it considers ($a_{d}\left(\lambda \right)$) and ($a_{CDOM}\left(\lambda \right)$) as an integrated variable called ($a_{dg}\left(\lambda \right)$).

The use of IOPs for the evaluation of phytoplankton blooms has been applied in recent years [14]. Santamaría-del-Ángel et al. (2015) [5] and Aguilar-Maldonado et al. (2018) [24] used in situ data as input to calculate an index, called the IOP index, which values could be associated to bloom or non-blooms conditions. Later, Aguilar-Maldonado et al. (2018) [20] adapted the in situ IOP index for using satellite products ($a_{phy}\left(\lambda \right)$ and $a_{dg}\left(\lambda \right)$) as input. The advantage of using satellite products was mapping the index at a moderate resolution scale on a specific moment. However, their approach was based on the evaluation of spatial anomalies, that is, the reference conditions were determined based on the spatial differences of these variables during some single time (day or month). The calculation of anomalies is used to determine the change magnitude necessary to assert that a specific phenomenon is causing an effect [25,26,27,28]. According to the anomalies theory, the baseline can be interpreted as the boundary on which if a value is above it is described as a positive anomaly (or increase), while if a value is below it indicates a negative anomaly (or decrement) [29]. Simple or climatological anomalies can be calculated. In climatological anomalies, the average value of the data is replaced by twelve monthly averages. Both simple anomalies and climatological anomalies are defined locally, their definition is valid only for the data series region. To be able to compare different regions is necessary to calculate the standardized anomalies, which are based on data standardization or Z transformation [29]. The concept of standardized anomaly is widely used in physical oceanography [30,31].

In this study, we propose the application of the IOP index, based on the use of time series standardized anomalies, to build a baseline of phytoplankton ($a_{phy{,}_{GIOP}}$) and detritus plus CDOM ($a_{dg{,}_{GIOP}}$) absorption coefficients obtained from satellite products. The objective is to evaluate deviations from the baseline that can be associated to bloom conditions according the IOP index results. We worked with data from Todos Santos Bay (TSB) (Baja California, México) and specifically with one bloom observed during May–June 2017 when in situ data were available from previous studies.

## 2. Methodology

### 2.1. Study Area and Field Data

TSB is a semi-enclosed bay located on the northwestern Pacific coast of Baja California (Mexico), approximately 100 km south of the Mexico–USA border (Figure 1). The bay is limited to the north by Punta San Miguel and to the south by Punta Banda. Surface water characteristics in this area are closely related to the California Current System (CCS), which produces the upwelling of cold and nutrient-rich subsurface water along the Baja California peninsula coast. The predominant circulation pattern is a southeastern flux into the bay, except when there is a change of flow direction to the northeast, Punta Banda, which induces surface water to flow out of the bay [32]. Primary productivity in TSB is characteristically high (daily average 1.03 g C m^−2^ inshore and 0.18 g C m^−2^ offshore) [33,34]. The city of Ensenada (228 km^2^, 466,815 inhabitants according to the last population census) [35] is located to the east of the bay. The Ensenada Harbor is one of the most important harbors in the Mexican Pacific. Aquaculture activities of high economic importance are carried out inside or near the bay, among them, tuna fattening and the cultivation of bivalve mollusks (such as oyster, clam, mussels, mule paw and ax callus).

TSB is characterized by a dominance of diatoms during the upwelling season (spring–summer), which alternates with a dominance of dinoflagellates when nutrients are depleted [36]. Recurrent dinoflagellate blooms have been observed since at least 1901 [32]. However, dinoflagellate algal bloom (DAB) events in this area have increased considerably in extension and frequency over the past two decades [37]. Aguilar-Maldonado et al. (2018) [24] reported a bloom during June 2017 that lasted three weeks. Field sampling was done on June 2, and surface water samples were obtained from the six stations represented in Figure 1. Water column depth is about 20 m in the field points located nearer the coast (4, 5 and 6). Water column depth increases to 40 m in field points 1 and 2, and it reaches 120 m in field point 3 [38]. The bloom was analyzed in Aguilar-Maldonado et al. (2018) [24], and their results are used in this study to compare field data with remote sensing data.

### 2.2. Image Processing and Calculations of the IOP Satellite Index

The methodology is based on the use of an extensive satellite database of the GIOP model of the TSB, in order to obtain the baseline. The variables $a_{phy{,}_{GIOP}}$ and $a_{dCDOM{,}_{GIOP}}$ were derived from images of the moderate resolution imaging spectroradiometer (MODIS)-Aqua sensor (https://oceancolor.gsfc.nasa.gov/atbd/giop/). We download daily Ocean Level-2 products, distributed by the NASA Goddard Space Flight Center’s Ocean Data Processing System (ODPS) which are already atmospherically corrected. The images spatial resolution was 1 km^2^. 3 × 3 pixels windows were taken around the pixel of interest, and the aggregation method of these pixels was the average. Absorption coefficients data were obtained using NASA’s SeaDAS V.7.5 software.

Two points were selected (points 4 and 6 in Figure 1) to extract their time series. These points were selected based on their strategic interest, because historical records of blooms events have been reported in the bivalve mollusk culture area and tuna fattening farms. To build the May and June baseline, daily images were used for the months of May and June since 2003 to 2016 (Table 1 summarizes the number of data available), and the following procedure was applied.

First, the absorption coefficients, $a_{phy{,}_{GIOP}}$ and $a_{dCDOM{,}_{GIOP}}$ for the pixels of interest, were standardized applying Equation (1):(1)$$Z=\frac{x-\bar{x}}{SD}$$
where: x is the value to be standardized (each day absorption coefficient);$\bar{x}$ is the average of the studied period (for May, all May data since 2003 to 2016; for June, all June data since 2003 to 2016);*SD* is the standard deviation of the studied period (for May, all May data since 2003 to 2016; for June, all June data since 2003 to 2016).

Second, the satellite IOP index was calculated as a standardized orthogonal empirical Function (SOEF) for each day, according to Equation (2):(2)$$IOP_{index\;satellite}=\left[\left(b_{1,1}*Za_{phy{,}_{GIOP}})+(b_{1,2}*Za_{dCDOM{,}_{GIOP}}\right)\right]$$
where $Za_{phy{,}_{GIOP}}$ and $Za_{dg{,}_{GIOP}}$ are the $a_{phy{,}_{GIOP}}$ and $a_{dCDOM{,}_{GIOP}}$ standardized values and $b_{1,1}$, $b_{1,2}$ are the first eigenvectors resulting from the SOEF, to see in detail the procedure see Aguilar-Maldonado et al. (2018) [20].

Then, the average satellite IOP index was calculated for May and June, and this will be considered the baseline for that specific months in that specific region. The baseline will be continuously feed each year with new data. Once we have calculated the baseline of satellite IOP index, we can compare any data with that baseline value.

The values of the satellite IOP index for May–June 2017 were calculated and classified as follows: (1) Values in the interval (−1, 1) indicate average values of the specific site or non-bloom conditions; (2) values in the interval (1, 1.6) are positive anomalies (above the average) and represent a transitional state from anomalous to average values, or vice versa (decaying or growing bloom conditions), depending on the behavior of the data set; (3) values higher than 1.6 are strongly anomalous and indicate a phytoplankton bloom event. These thresholds are defined thanks to the standardization of the data. In a normal distribution, the inverse cumulative distribution function (ICDF) [5], defines 1.6 standard deviations as the limit of values without noise with a 90% confidence level.

The satellite IOP index results were compared with the field IOP index results obtained by Aguilar-Maldonado et al. (2018) [20] for the same data and points. A Spearman rank correlation analysis was performed to test statistically significant correlations between the satellite and the field data absorption coefficients. Daily satellite chlorophyll *a* concentration (mg·m^−3^) was obtained from MODIS Aqua, MODIS Terra and VIIRS (Visible Infrared Imaging Radiometer Suite), with the OCI algorithm. A multi-sensor approach was used to overcome cloud cover. These data were used to compare May and June data in Todos Santos Bay with the satellite IOP index.

## 3. Results and Discussion

The calculated baseline of satellite IOP index, for the period since 2003 to 2016, for field point 4 was 0.001 in May and 0.006 in June, and for field point 6 it was 0.002 in May and 0.009 in June. As time goes by the data base of $a_{phy{,}_{GIOP}}$ and $a_{dCDOM{,}_{GIOP}}$ will be increasing with new data, and consequently the baseline can be modified. There exist studies which warn that climate change can have an effect on phytoplankton due to changing water column stratification and resource availability, mainly nutrients and light, or intensified grazing by heterotrophs [39,40]. So, baseline data feeding is of paramount importance because phytoplankton blooms characteristics (timing, frequency, composition and intensity) are expected to change with climate in a way that is hard to predict [41]. Already, several studies have observed advance in phytoplankton spring bloom timing and changing bloom magnitudes [40]. To be able to distinguish a bloom event, daily satellite IOP index must be compared with this baseline. In Table 2 the IOP index statistics from 2003 to 2016 are summarized. The frequency of IOP index values above 1.6, and thus in active bloom conditions is between 6% to 7% for that period. In Table 3 we analyze the monthly frequency of IOP index values >1.6 by year. The temporal series show a trend inversion since 2010–2011. From 2003 to 2009, the satellite IOP index >1.6 frequency was between 8% and 38% (except for some months it was 0%). From 2010 to 2016, the IOP index did not exceed 1.6, except in June at point 6 (11%). Even in the months with higher frequency of IOP index >1.6, the persistence of these values was reduced mostly to one observation (a day). Only sporadically we detected active blooms condition during two (May 2005 point 4, and May 2003 point 6) or three (May 2006 and June 2007 at point 4) consecutive days.

The availability of long term data series is essential to distinguish the natural variability of the ecosystem from other causes of variability such as climatic variability or anthropogenic factors [4,5]. In the city of Ensenada, they operate six wastewater treatment plants that provide secondary treatment to the water that collects the sewage system, with a total installed capacity of 917 L/s and an estimated treated flow of 552 L/s [42]. Two of this plants started to operate one in 2009 “Northeast plant” (52 L/s) and the other in 2010 “Maneadero plant” (7.2 L/s) (Figure 1). Water quality improvement in the receiving waters, have been previously observed in Todos Santos Bay after construction of new treatment facilities [43]. Treated wastewater discharges have to accomplish the Mexican government pollution thresholds for wastewaters [44]. This change can affect phytoplankton biomass and other optically active components (detritus and CDOM), decreasing their concentration. In addition, the cultivation of bivalve mollusks in the bay (see area in Figure 1) has increased in recent years [45]. Bivalve filter-feeding mollusks are important components of coastal ecosystems because they remove large quantities of suspended material from the water, primarily phytoplankton [46]. Both factors, wastewater treatment and bivalve aquaculture could have contributed to a decrease in the satellite IOP index.

The daily temporal evolution of the satellite IOP index from 1 May to 30 June 2017, at points 4 and 6, is shown in Figure 2 and Figure 3. The bars of the satellite IOP index were colored to facilitate interpretation as follows: (1) Green color for values in the interval (−1, 1) show the average values of the specific site or non-bloom conditions; (2) yellow color for values in the interval (1, 1.6) are above the average and represent a transition state, and can be considered as a bloom pre-alert situation, and (3) red color for values higher than 1.6 are highly anomalous and indicate an active phytoplankton bloom, and can be considered as a bloom alert situation. Satellite chlorophyll *a* concentration (mg·m^−3^) is overlapped in gray color bars.

Daily data available for field point 6 is more reduced than for field point 4 (13 days as compared to 23 days). There was no satellite information available for the day the field sampling was conducted (2 June 2017) at none of the two points. In a conservative approach, in the absence of satellite images it should be considered that the worst scenery remains until new information could be processed. In field point 6 (Figure 3) a bloom pre-alert situation (yellow color) started on May 24 and was kept until 8 June 2017, as there was no satellite information during the 14 days the pre-alert was kept. On June 2 there was an active bloom of a dinoflagellate, *Lingulodinium polyedrum*; cell counts exceeded 40 thousand cells L^−1^ [24]. For this reason, it is important to clearly define which management measures should be adopted during bloom pre-alert. On June 10 there was a decrease in the satellite IOP index that reached average values or non-bloom conditions, but this lasted only until June 14 when an active bloom condition was detected. At field point 4, the prior warning situation was reached several times, but it was expanded to a maximum of 4 days because the availability of satellite data was greater and the new data confirmed the decrease in the value of the satellite IOP index to non-bloom conditions. TSB is a highly cloudy area which prevents the availability of daily satellite images. In the example, the maximum period without satellite information was 14 days. This is a relevant challenge for using polar-orbiting satellites, such as MODIS, that has been already pointed out in other cloudy areas such as Cheasepeake Bay [9]. To improve this temporal resolution, two alternatives arise, either using a multi-sensor approach or using sensors on geostationary platforms [8,9].

To assess the utility of the proposed methodology, we compared IOP index values with satellite chlorophyll *a* values which is a more traditional approach to define high biomass phytoplankton blooms. In Figure 2 and Figure 3 we observed that at field points 4 and 6, chlorophyll *a* concentration (mg·m^−3^) is, in general, below 24 mg·m^−3^ during non-bloom conditions, and during bloom conditions it rises above this level. In our work, we tried to emphasize the need to compare one area with its own long-term values (baseline) to be able to detect bloom conditions. There is not a unique biomass value that can be considered as the threshold between bloom or non-bloom, because different areas have different characteristics and different levels of biomass can be considered a bloom. But, we could define a phytoplankton bloom as a deviation to higher values than standard seasonal patterns at a specific site. The advantage of this methodology is that the baseline is calculated for any specific site to detect deviations from that baseline, and that makes possible the generalization of the methodology to any area, and to all trophic conditions. Previously, the IOP index has been tested in optically complex waters with different conditions, including the Upper Gulf of California which is considered as one of the most biologically productive marine regions (peak chlorophyll a concentrations of 18.2 mg m^−3^) [24]. Even in areas with extremely frequent blooms (Campeche, Yucatan Peninsula) the IOP index is able to distinguish active blooms from decaying blooms [20]. The advantage of the IOP index is that it is calculated having taken into account the three components of optically complex waters (phytoplankton, CDOM and detritus), so it is able to distinguish between increases in phytoplankton biomass from increases in CDOM (due to the degradation of organic matter, that is, the degradation of the dead phytoplankton that formed the bloom) or detritus (from terrestrial discharges).

Having taken into account the calculation process of the satellite IOP index, the relative weight of $a_{dCDOM{,}_{GIOP}}$ is lower than the $a_{phy{,}_{GIOP}}$. In consequence, to get high values of the IOP index (close to 1.6) as a consequence of the increase in CDOM and detritus, this increase has to be striking. As an example, the IOP index value suggests a pre-bloom situation at point 4 on June 8 (Figure 2), when the chlorophyll concentration is relatively low. This can be due to a very anomalous increase in CDOM. The degradation of phytoplankton cells from previous blooms causes the increase in CDOM. The morphology and dominant currents of TSB favor a mechanical accumulation of this CDOM in the area of field points 4 and 6. This is very important because sometimes water discoloration is wrongly attributed to phytoplankton blooms, when in fact the bloom is not active, and even may have originated in a near area and been displaced by currents. Unfortunately, we did not have images available for the 10 days prior to June 8, due to cloud cover, to be able to unveil our hypothesis of a previous bloom.

We analyzed the contribution of each absorption coefficient ($a_{phy{,}_{GIOP}}$ and $a_{dCDOM{,}_{GIOP}}$) to total absorption from 2003 to 2016. The average contribution of phytoplankton was 56.4% for IOP index values below 1; 45.4% for IOP index values between 1 and 1.6; and 57.0% for IOP index values above 1.6. That is, phytoplankton contribution is higher in non-bloom and in active bloom conditions, while detritus and CDOM are higher in decaying bloom conditions. In Figure 4, we map the satellite absorption coefficients from 25 May to 10 June 2017 in Todos Santos Bay. In the 2017 bloom, we observed a 63% contribution of $a_{phy{,}_{GIOP}}$ in field point 6, when active bloom conditions were detected. This higher contribution of phytoplankton absorption can be well observed in Figure 4.

Satellite absorption coefficient were compared to previously published field data. The Spearman correlation analysis between absorption coefficients showed a statistically significant positive correlation (*α* = 0.05, n = 6). The correlation coefficient between $a_{phy\;field\;data}$ and $a_{phy\;satellite}$ was 0.809 (*α* < 0.05), and correlation between $a_{dCDOM\;field\;data}$ and $a_{dCDOM\;satellite}$ was 0.671 (*α* < 0.05). Thus, the correlation was higher for phytoplankton absorption coefficient, but in both cases can be considered a good fit [7,47,48]. The satellite IOP index and the field IOP index behavior was similar in the six stations sampled on June 2017 (Figure 5). In general, it can be observed that the satellite IOP index shows higher values than the field IOP index, but the classification of the bloom condition was the same for all the field points, except for field point 5. The field IOP index classified this point as non-bloom conditions, while the satellite IOP index classified it as in active bloom conditions. Field points 1, 2, 3 and 4 were in non-bloom conditions, and had average values for this region. Field point 6 was in active bloom conditions, it showed the highest anomaly with an IOP index value above 1.6.

The difference between the field IOP index and the satellite IOP index may be due to several factors. The first is temporality, field observations are rarely done simultaneously with remote sensing [14]. In this study, samples were taken on 2 June 2017, while satellite data were a composite of the available images since 25 May to 10 June 2017. In the specific case of field point 5, there were only three images available, less than for the other field points. Laboratory results are based on discrete water samples of a limited volume, while satellite results represent the average value of one pixel (1 km^2^) [14]. In addition, it is known that phytoplankton blooms are characterized by a patchy distribution [10].

The Mexican Bivalve Mollusks Health Program aims at creating a historical database in order to have background information of phytoplankton species of the Mexican littoral [49]. For this purpose, a Permanent and Systematic Sampling Program has been defined. A weekly sampling frequency and analysis of seawater, for phytoplankton cell count, was established in previously determined sampling stations, under non-bloom conditions. The quantitative analysis of collected water samples can offer information at species level but it has some limitations. The first limitation is that is not possible to have quantitative information always due to limited laboratory facilities and personal, so in that case qualitative analysis is done, with the consequent loss of information [49]. The second one is the complexity of having high quality data of quantitative phytoplankton analysis [36,50]. It is common a significant variability among laboratories in the methods used to sample, preserve and count, identify and measure phytoplankton cell volume, the taxonomic nomenclature used and resolution reported. The satellite IOP index methodology proposed in this study aims to be a complementary tool for permanent on-site monitoring programs. This methodology does not provide information at species level, but allows to have a complete spatial coverage of TSB (or any other coastal area), while on-site sampling should be limited to previously fixed sampling stations. This can be very helpful to determine the extent of blooms events and affected areas, and also may detect bloom events in non-monitored areas. Blooms can occur even at extremely low levels of Chla in oligotrophic seas (<0.15 mg m^−3^) [51,52], without noticeable changes in water color due to low cell density which complicates their detection [8,10], but the use of IOPs combined with the index approach proposed in this work is able to detect any deviations from average values. One of the objectives of permanent monitoring programs is to obtain background information of phytoplankton. In this sense, the use of the satellite IOP index provides reliable information that may be more objective than traditional phytoplankton analysis. It is true that this method cannot offer taxonomic information at present state, but it provides important information about the background levels of phytoplankton, detritus and CDOM that can be very useful for detecting anomalies. Therefore, it can serve as a first-level permanent monitoring tool to support decisions about when and where it is necessary to take in situ samples.

Regarding sampling frequency, the Permanent Sampling Program [49] is quite ambitious in his principles, it purposes weekly frequency for phytoplankton quantitative analysis, but this is not easy to accomplish. MODIS GIOP products, which are the base for calculating the satellite IOP index, are available at a daily frequency. However, this frequency could be reduced due to cloud cover. Currently bloom events are not well documented and resolved because the mismatch in the time scales of phytoplankton biomass variability (days-weeks) and in situ sampling (weeks months). In this study, the TSB was selected to test the methodology and, despite being a particularly cloudy area, the satellite IOP index proved to be useful and had a higher frequency than in situ sampling. Therefore, it is expected that in less cloudy areas temporal resolution will be improved. Additionally, a multi-sensor approach or the use of sensors on geostationary platforms can increase temporal resolution.

## 4. Conclusions

Long term monitoring is necessary to define the baseline conditions of a specific variable in a specific area. In situ marine monitoring programs are used to compile background information on phytoplankton abundance, which is especially important in coastal areas with uses such as aquaculture. However, field sampling must be limited to specific points and data due to cost-effort efficiency. The use of remote sensing products is useful to complete these programs because it has the advantage of being able to map entire areas with high frequency. In this research, two points have been selected to show how the analysis of absorption coefficients and the satellite IOP index baseline allow detecting trend changes. In addition, the satellite IOP index allows detecting active phytoplankton bloom conditions thanks to the anomalies theory. The satellite IOP index can be used as a first-level permanent monitoring tool to support decisions about when and where it is necessary to take in situ samples, and to define a strong background to detect anomalies.

## Figures and Tables

**Figure 1 sensors-19-03339-f001:**
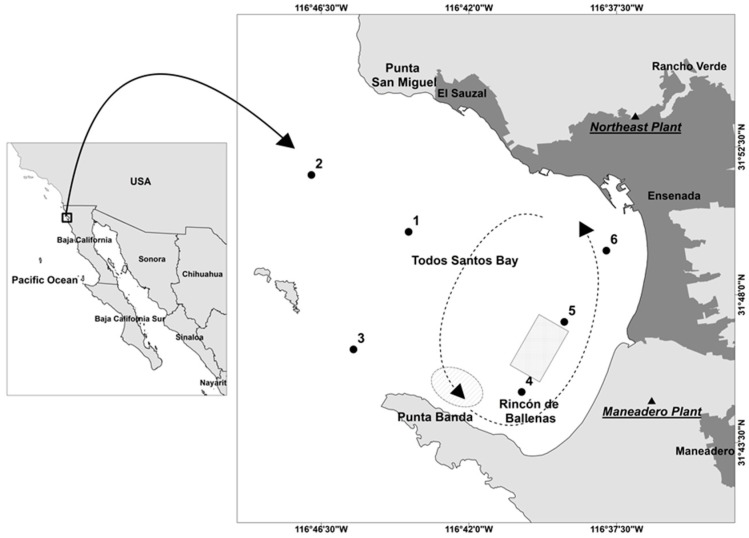
Study area, Todos Santos Bay (Baja California, Mexico). Points from 1 to 6 are field stations. The rectangle shows a bivalve mollusks cultivation area and the oval a tuna fattening area. Dashed lines and arrows indicate predominant circulation pattern.

**Figure 2 sensors-19-03339-f002:**
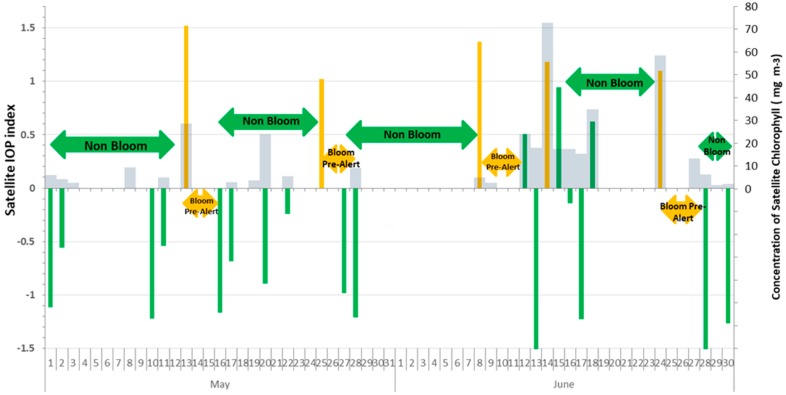
Temporal evolution of the satellite IOP index and satellite chlorophyll *a* (mg m^−3^) from 1 May to 30 June 2017 at field point 4. Green color bars show IOP index in non-bloom conditions; yellow color bars show IOP index in bloom pre-alert conditions, gray bars are chlorophyll *a*. Absence of bars is due to cloud cover.

**Figure 3 sensors-19-03339-f003:**
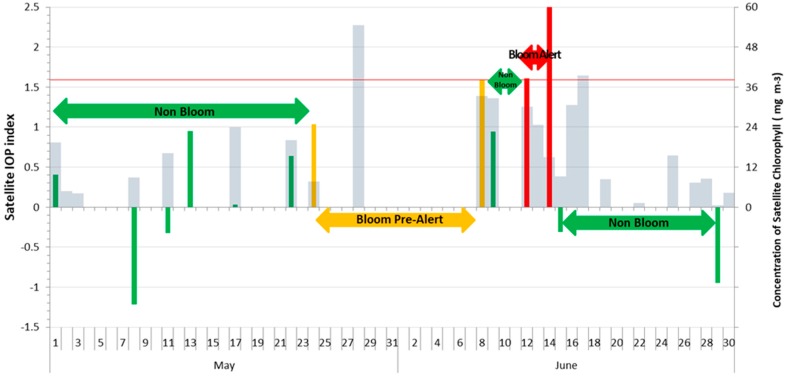
Temporal evolution of the satellite IOP index and satellite chlorophyll *a* (mg m^−3^) from 1 May to 30 June 2017 at field point 6. The red line defines the limit of anomalous conditions (>1.6 standard deviations of IOP index) and active bloom conditions. Green color bars show IOP index in non-bloom conditions; yellow color bars show IOP index in bloom pre-alert conditions and red color bars show IOP index in active phytoplankton bloom; and gray bars are chlorophyll *a*. Absence of bars is due to cloud cover.

**Figure 4 sensors-19-03339-f004:**
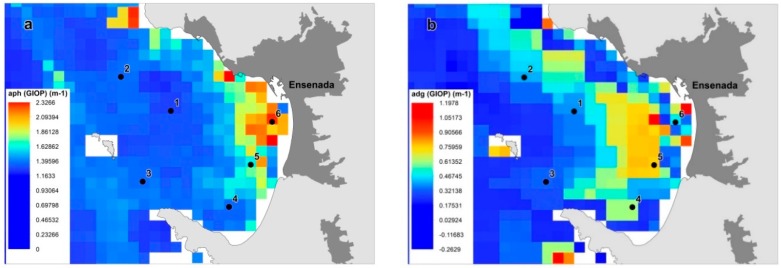
(**a**) $a_{phy\;satellite}$ and (**b**) $a_{dCDOM\;satellite}$ from 25 May to 10 June 2017 in Todos Santos Bay.

**Figure 5 sensors-19-03339-f005:**
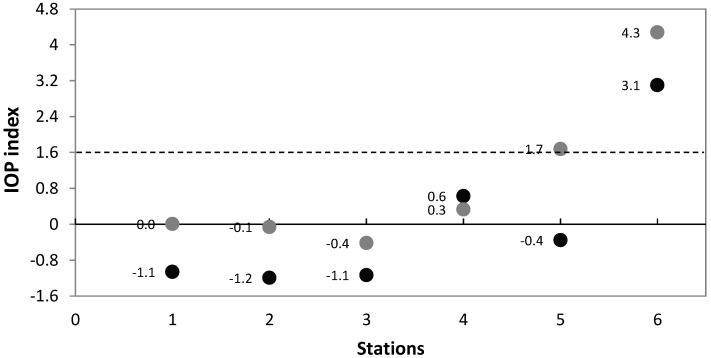
IOP index results in Todos Santos Bay. The results of the IOP index with field data obtained on 2 June 2017 are represented graphically with black dots and the results of the satellite IOP index from 25 May to 10 June 2017 are represented by gray points. The points on or above the dashed line are in active bloom conditions.

**Table 1 sensors-19-03339-t001:** Available remote sensing data used to build the baseline since 2003 to 2016.

Point	Month	# Observed Days (2003–2016)
4	May	111
	June	146
6	May	101
	June	115

**Table 2 sensors-19-03339-t002:** Inherent optical properties (IOP) index statistics from 2003 to 2016. Frequency of IOP index in each interval is calculated as number of cases divided by number of total observations.

	Frequency of IOP Index Values (%)			Minimum IOP Index	Maximum IOP Index
	<1	1–1.6	>1.6		
Point 4 May	81	13	6	−1.31	5.20
Point 4 June	85	8	7	−0.88	5.16
Point 6 May	79	15	6	−1.24	3.54
Point 6 June	87	7	6	−1.12	4.29

**Table 3 sensors-19-03339-t003:** Monthly frequency (%) of IOP index values >1.6 by year from 2003 to 2016. Frequency of IOP index (%) is calculated as number of cases divided by number of month observations.

	Point 4 May	Point 4 June	Point 6 May	Point 6 June
2003	0	17	38	0
2004	8	18	9	25
2005	25	11	17	20
2006	29	10	0	0
2007	0	27	0	14
2008	11	0	0	0
2009	0	20	25	0
2010	0	0	0	0
2011	0	0	0	11
2012	0	0	0	0
2013	0	0	0	0
2014	0	0	0	0
2015	0	0	0	0
2016	0	0	0	0

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
