# Peer review of "Detection of Phytoplankton Temporal Anomalies Based on Satellite Inherent Optical Properties: A Tool for Monitoring Phytoplankton Blooms"

_sensors, 2019, doi:10.3390/s19153339_

Round 1

Reviewer 1 Report

The manuscript entitle “Detection of phytoplankton temporal anomalies based on satellite Inherent Optical Properties: a tool for monitoring phytoplankton blooms” aims to identify algal blooms events as support to programs of algal bloom monitoring. Overall, the document is interesting and very well-written. My unique concern is about a possible regional limitation of method proposed. It is not clear if it will be possible to create a suitable baseline for eutrophic waters, where chlorophyll-a is extremely high and algal blooms are frequent. It would be interesting to insert a discussion about this for the audience. In addition, the model should be applied in some MODIS images, showing the spatial distribution of non-algal and algal bloom events. Specific observations were pointed out.

Specific comments

Line 53. Remove comma after et al.

Lines 73-75. Quite pretentious! For years, the study has been applied on the use of IOPs in algae bloom cases.

Line 107. Km2?

Line 108. Include “)” after census.

Lines 164 and 166. Use dot comma instead of only comma.

Line 242. Include final dot.

Line 269. Km2.

Line 293. “avoid missing bloom event” depends on the availability of images and according to the document TSB is cloudy.

Author Response

Dear reviewer,

Best regards

Reviewer 2 Report

This manuscript describes the use of the Inherent Optical Properties (IOP) calculated from satellite imaginary to determine the occurrence of phytoplankton blooms in the Pacific coast of Baja California. The detection method is based on IOP deviations from a baseline calculated with daily satellite images of 2003-2016. The performance of the method is contrasted with field data obtained in 2017. The manuscript addresses on a relevant topic of paramount interest from the viewpoint of the coastal management. The work is fairly well written and the results are clearly presented. However, I think that the manuscript would be improved in several points:

(1) As indicated by the authors, IOP depends on both light absorption coefficient of the phytoplankton and presence of detritus and light absorbing dissolved substances in the water column. Multiple articles demonstrate that the phytoplankton light absorption coefficient depends not only on its biomass but also on physiological features of the own community that in turn are determined by its photo-acclimation and nutrient status, pigment cell content and/or cell size. Therefore, as occur with chlorophyll, I am also not sure that IOP is an ideal indicator of phytoplankton biomass or growth. In this context, I would expect that the authors devote part of the results and discussion to support the hypothesis that deviations from satellite IOP baseline are effectively indicating occurrence of phytoplankton blooms (i.e. rapid increase in the phytoplankton biomass) instead of being due to shifts in the community composition or any other factor non-necessarily related to phytoplankton (for instance, changes in macro-benthos communities, terrestrial discharges of dissolved and/or particulate matter, etc.).

(2) The authors should deal with the uncertainties of satellite reflectance to infer IOP (e.g. see the article published in PiO in 2018; 160: 186-212). I recognize that the performance of the satellite is researched (lines 246-249 and Figure 6). However, it is clearly insufficient given that the correlations are based only on data obtained in six stations at a given date; consequently, it indicates that satellite IOP reproduces horizontal variability of field IOP, but it does not mean necessarily that the temporal variability is similarly reproduced. Furthermore, IOP index averaged from May 25 to June 10 is compared with IOP index obtained on 2 June from field data. Taking into account that the phytoplankton blooms are normally rapid processes lasting hours or few days, I cannot see the sense of this comparison.

(3) Related to the previous point, I do not understand the utility of the information shown in Figures 2 and 5 based on monthly means. Probably it would be more useful to provide with a statistical analysis of the time series. For instance, it would be interesting to show which is the frequency of daily IOP index values above 1.6 and which is their persistence (related to the duration of the bloom); which is the own variability range of the index, how the relative contribution of phytoplankton and detritus plus CDOM changes throughout the time and during a bloom. In my opinion, there is a good data base (Table 1) that can be used to support the proposed method, but the authors did a poor analysis of it.

(4) As far as I know, a phytoplankton bloom in coastal oceanography is defined as the process forwhich phytoplankton abundance exceeds certain value. I would be in agreement with a definition of phytoplankton bloom based on a rapid change in the optical properties of the water column attributable to phytoplankton growth for a given area; however, a relationship between thresholds of IOP index and biomass or abundance should be defined for that area. In other words, I will expect that the authors indicate the 'meaning' of their threshold of IOP index in terms of biomass or abundance of phytoplankton (or chlorophyll a concentration). Similarly, I think that the relationship between the satellite IOP index and any other classical methods for determining the bloom occurrence should be researched in this manuscript. Otherwise, it is difficult to assess the utility of the method.

(5) I miss a satellite image of the horizontal distribution of the measured properties in the study area (absorption coefficients), just to give an idea of their magnitude and illustrate the representativeness of the selected stations.

Other minor points:

(6) Lines 105-107. Some figures of primary production for TSB should be shown in the text to guide the readers regarding to what it is meant with "high productivity".

(7) Lines 134-139. The resolution of the satellite images is 1 Km2. Consequently, the method used to select or interpolate spatially the pixels relevant for each station must be described.

(8) Lines 162-168. I wonder if values exceeding 1.6 times the IOP index baseline represent necessarily the production of a bloom (see also my point #4). Please, to justify.

(9) Line 182. Please, to be more explicit regarding to how it is expected that climate changes the characteristics of the blooms.

(10) Line 185. Some of deviation measurements should be shown in the graphs.

(11) Line 195. This improvement in the water quality should supported with data. Otherwise, it appears speculative.

(12) Line 226. These field data must be shown (not only commented).

(13) Lines 271-285. This section of the paragraph can be shortened.

(14) Lines 294-295. These lines reinforce the necessity of providing with a quantitative definition of bloom (in terms of chlorophyll or biomass) in the study area.

Author Response

Dear reviewer,

Best regards

Round 2

Reviewer 2 Report

I thank the authors for considering all my comments and suggestions and modifying the manuscript accordingly. I think that the authors have done a rather valuable work of revision and that the manuscript has improved substantially. In particular, in the new version, the time series of IOP index is numerically analyzed (new Tables 2 and 3), data of chlorophyll concentration are shown (Fig. 2 and 3) and the spatial distribution of optical properties is included (Fig. 4). Furthermore, some missed information in the previous version is shown and modifications of the text have been done throughout the manuscript in order to clarify some issues.

I reinforce my view that the manuscript is useful for publication, although I have still some minor comments that the authors might consider before sending the final version.

(1) The shown chlorophyll data are quite helpful to understand the utility and limitations of the IOP index. However, I think that the authors made limited use of the comparison between chlorophyll and IOP index to do a more substantial criticism to the proposed method. For instance, the new data indicate that in some occasions the IOP index value suggests pre-bloom situation even although the chlorophyll concentration is relatively low (8th June, point 4). Furthermore, during the bloom of 12-14 June in the point 6, the chlorophyll concentration is not excessively elevated; in fact, chlorophyll appears to decrease while IOP index increased. These data support the suggestion of the authors that phytoplankton light absorption only partially counted for the variations of the IOP index (lines 281-286). I would be in agreement with variation in the IOP index would be due to production of CDOM from the pre-formed phytoplankton biomass. However, it implies that the explosive growth of the phytoplankton biomass (which characterizes a bloom; otherwise, we are talking about other phenomenon) and the consequent changes in the IOP index would be non-synchronized. Furthermore, under determined conditions, CDOM from phytoplankton (or other plankton groups) would accumulate. Obviously, this has implications about how and when during the development of a bloom the usage of the IOP index would be more or less adequate, but I cannot see comments in the discussion focusing on this issue.

(2) Line 123. It would be informative to indicate the depths of the stations.

(3) Line 133. I think that it should be squared km.

(4) Line 289. I guess that 0.809 and 0.671 are correlation coefficients (please, clarify).

(5) Line 301. Units should be indicated in the graph or legend.

(6) Lines 320-322. The sentence commencing with “The first one…” is confused to me. Please, rewrite.
